# A Generative Adversarial Network with Spatial Attention Mechanism for Building Structure Inference Based on Unmanned Aerial Vehicle Remote Sensing Images

**Hao Chen** [1,*], **Zhixiang Guo** [1], **Xing Meng** [2] **and Fachuan He** [1]

1    School of Electronics and Information Engineering, Harbin Institute of Technology, Harbin 150006, China; 22s005075@stu.hit.edu.cn (Z.G.); 20b905001@stu.hit.edu.cn (F.H.)
2    Institute of Defense Engineering, Academy of Military Sciences, Beijing 100036, China; mengxthu@126.com
*    Correspondence: hit_hao@hit.edu.cn

**Abstract:** The acquisition of building structures has broad applications across various fields. However, existing methods for inferring building structures predominantly depend on manual expertise, lacking sufficient automation. To tackle this challenge, we propose a building structure inference network that utilizes UAV remote sensing images, with the PIX2PIX network serving as the foundational framework. We enhance the generator by incorporating an additive attention module that performs multi-scale feature fusion, enabling the combination of features from diverse spatial resolutions of the feature map. This modification enhances the model's capability to emphasize global relationships during the mapping process. To ensure the completeness of line elements in the generator's output, we design a novel loss function based on the Hough transform. A line penalty term is introduced that transforms the output of the generator and ground truth to the Hough domain due to the original loss function's inability to effectively constrain the completeness of straight-line elements in the generated results in the spatial domain. A dataset of the appearance features obtained from UAV remote sensing images and the internal floor plan structure is made. Using UAV remote sensing images of multi-story residential buildings, high-rise residential buildings, and office buildings as test collections, the experimental results show that our method has better performance in inferring a room's layout and the locations of load-bearing columns, achieving an average improvement of 11.2% and 21.1% over PIX2PIX in terms of the IoU and RMSE, respectively.

**Keywords:** building structure inference; mapping relationship; additive attention module; line penalty term; UAV remote sensing image

## 1. Introduction

The building structure plays a crucial role in applications such as building strength assessment and controlled demolition [1–4]. In certain situations, such as with very old buildings [5] or buildings that cannot be entered, often their architectural drawings are missing, or it is not possible to directly obtain information about the building's structure. In these situations, practitioners may manually infer the building's structure from the building's outward appearance. With the development of remote sensing technology, unmanned aerial vehicles (UAVs) with high-resolution images have been widely used in different fields [6–9], making it easier to obtain the appearance features of buildings, which makes it possible to use UAV images to infer the structure of buildings.

Currently, there is relatively little research on inferring building structures. More attention is focused on the semantic segmentation and recognition of building floor plans, which involves extracting more information about the rooms, such as their categories and areas, based on the structural form of the building. Such methods involve designing a specific set of rules to process the input building floor plan. Traditional methods mainly use line detection to determine the boundaries of the building and then recognize the

contours of the rooms based on different rules. Macé [10] used a combination of Hough transforms and image vectorization for the line detection of houses and then used the polygonal approximation convex decomposition method to divide and extract the rooms based on the extracted results. Ahmed et al. [11] processed building floor plans using graphic and text segmentation methods, classified wall thickness, and achieved the precise segmentation of building floor plans to extract houses. De las Heras [12] established wall hypothesis rules to model parallel line elements in building floor plans, deduced wall features based on probability, and extracted walls. Additionally, introducing topological relationships in traditional methods can also help recognize the contours of building floor plans. Gimenez et al. [13] formulated recognition rules for building outlines based on the topological relationships between points and used a heuristic algorithm to identify the discontinuous parts of walls in 2D planes, generating 3D models that comply with the industry rules. Jang [14] preprocessed planar building maps, separated semantic symbols from the map, and then performed vectorization operations on the map to find the topological relationship of the map and used the adjacency matrix method to obtain the relationship of the building structure skeleton. Methods that reduce data complexity, such as distribution analysis [15], have brought new research perspectives to the semantic recognition and segmentation of architectural floor plans.

With the in-depth research of deep learning methods, the research on architectural structures mainly includes two parts: semantic recognition and the segmentation of architectural floor plans, as well as the automatic generation of architectural floor plans. In the field of semantic recognition and segmentation, deep learning methods are data-driven and can improve the robustness of the method in extracting rooms, which helps researchers decrease the difficulties of designing extraction rules. Dodge et al. [16] used a fully convolutional network (FCN) for wall segmentation and used the faster R-CNN network to detect six types of objects (doors, sliding doors, gas stoves, bathtubs, sinks, and toilets) in building floor plans. Then, OCR was used to extract numbers from the floor plan to calculate the area of the house. Liu et al. [17] proposed the raster-to-vector algorithm, which uses a CNN-based vectorization operation to extract the outline structure of the building and then extracts deep constraints through the network. Lee et al. [18] proposed the Roomnet, which performs convolutional encoding–decoding operations on input house photos, extracts a series of layout key points such as corners, beams, and columns from the image, and then connects them in a logical order to obtain a house layout. Huang et al. [19] used a Generative Adversarial Network (GAN) to generate building floor plans with different color markings for different rooms as the input, and both the generator and discriminator were trained using CNN networks to recognize and generate floor plans. In the same year, Yamasaki [20] proposed the Maximum Common Subgraph (MCS) algorithm based on the idea of semantic segmentation. The algorithm extracts vertices from the graph in the form of an adjacency matrix, rearranges duplicate vertices through matrix multiplication to determine the best MCS, and then uses the FCN network to train and learn the segmented building floor plan. In the field of automatic generation, computer-aided automatic spatial layout design methods have been proposed in the field of architecture for a long time [21–23], with the main goal of generating spatial layouts of buildings based on certain constraints. With the proposal of the Generative Adversarial Network (GAN) [24] and the research of related methods, more data-driven spatial layout design networks have been proposed, such as House-GAN [25], Building-GAN [26], House-GAN++ [27], and ESGAN [28], which utilize generative adversarial networks to extract regularities and automatically design spatial layouts based on constraint conditions.

From the above methods, we can see that it is challenging to use appearance features to infer building structures. The building structure inference process can be seen as an image-to-image translation problem that needs to establish a mapping relationship between appearance features and the internal structure of the buildings. Based on the fact that UAV remote sensing images contain rich appearance features, a specialized database is established. The external features of buildings such as the distribution of doors and

windows as well as the location of entrances are used to build input images, and the ground truth is established according to the real CAD floor plan. By learning the mapping relationship between the two using a generator, the structure of the building can be inferred from the UAV remote sensing images. With the widespread application of deep learning in image processing, especially with the use of a PIX2PIX network [29], it is possible to establish a mapping relationship between images, which can be beneficial for inferring the structure of buildings from its appearance features. As a result, we chose PIX2PIX as the framework for our method. For establishing the mapping relationship, the generator of PIX2PIX focuses on extracting local features and does not effectively extract global features such as symmetry in building structures. To improve this limitation, an additive attention module is added to the generator that takes in the downsampling and upsampling feature maps from different layers and generates an attention map to replace the downsampling layers in the original generator for concatenation. By fusing multi-scale features, the module achieves the goal of extracting the global features of building structures. At the same time, we designed a novel loss function to ensure the completeness of line elements in the generated results. The loss function of PIX2PIX can effectively enhance the high-frequency information of the results, improving their resolution. However, it does not consider the relationship between pixels at different positions. Therefore, in our method, the existing loss function of the PIX2PIX network may result in incomplete straight-line features, such as walls, in the generated images. To better characterize the completeness of line elements, the output of the generator as well as the ground truth are transformed into the Hough domain, where the number of intersections is used to represent the completeness of the lines. By minimizing the loss function, we make the two images as close as possible, ensuring the completeness of the line elements in the images.

In summary, the following are our contributions:

1. In response to the limited research on inferring internal building floor structures using UAV remote sensing images, we propose an architecture inference network based on a PIX2PIX network backbone. The network takes the building appearance outline images as input and utilizes a trained generator for inference, achieving the specific task of inferring building structures.

2. In response to the characteristics of the dataset for this task, we introduce an adaptive attention module into the network. The inclusion of this module enhances feature extraction in the interested region, thereby avoiding the problem of global feature loss during downsampling to some extent and improving the accuracy of inference. At the same time, the introduction of the spatial attention mechanism improves computational efficiency and saves computing resources.

3. In order to address the issue of the original loss function's inability to effectively constrain the integrity of the results, we design a dedicated loss function that includes a penalty term for wall integrity in addition to the original loss function. Through measuring the number of intersection points after transformation in the Hough domain, this loss function can effectively constrain the results, giving sufficient attention to the integrity of the lines and walls, thereby improving the integrity of the walls in the results and increasing the accuracy of the inference.

## 2. Backgrounds

The Generative Adversarial Network (GAN), proposed based on the zero-sum game theory, consists of a generator and a discriminator, both of which are deep learning networks. The generator is trained to map the noise vector ($z$) to an output ($y$), as shown in Equation (1):

$$G : z \rightarrow y. \tag{1}$$

The objective of the generator is to make the data distribution of the output, $p_Y(y)$, approximate the distribution of real data, $p_X(x)$, as closely as possible. The discriminator is used to determine the authenticity of the generated output ($y$). The goal of a GAN is to train the generator such that the discriminator cannot distinguish between the generated

output and real data. To achieve this objective, it is necessary to find the optimal mapping function, as is given in Equation (2):

$$G^* = \underset{G}{\arg\min}\ \underset{D}{\max}\ L_{GAN}(G, D), \tag{2}$$

where $L_{GAN}(G, D)$ is the objective function of the GAN, as shown in Equation (3):

$$L_{GAN}(G, D) = \mathbb{E}_{x \sim p_x(x)}[\log D(x)] + \mathbb{E}_{z \sim p_z(z)}[\log(1 - D(G(z)))]. \tag{3}$$

where $x$ is the real data, $p_x(x)$ is the probability distribution of real data, $z$ is the random noise, and $p_z(z)$ is the probability distribution of noise data.

During the training process of the GAN, each training round consists of two parts: discriminator training and generator training. In the discriminator training phase, the generator parameters are fixed, and the discriminator is trained using both the generated results and real data to maximize the probability of a correct classification and distinguish the generated results from the real ones as much as possible. In the generator training phase, the discriminator parameters are fixed, and the generator is trained using generated data to make it as close to the real data as possible. These two phases are alternated until an equilibrium point is reached where the generator can generate realistic data and the discriminator can accurately distinguish between the real and generated results.

Based on the GAN proposed by Goodfellow, Isola proposed a general solution to the image-to-image translation problem using GANs known as the PIX2PIX network [24,29]. To obtain the mapping relationship between images, this network takes a pair of images as the input, with one image being used as the input to the generator and the other one being used as the real result, which is input together with the generator's output to the discriminator for discrimination, thus completing the training process. The network combines the distance loss function and the loss function of the generative adversarial network to improve the quality of the output image, as shown in Equation (4):

$$G^* = \underset{G}{\arg\min}\ \underset{D}{\max}\ L_{cGAN}(G, D) + \lambda L_d(G), \tag{4}$$

where $L_{cGAN}(G, D)$ is the loss function of the conditional GAN [30], as shown in Equation (5). $L_d(G)$ is the $L1$ distance loss function [31], as shown in Equation (6). $\lambda$ is the weight of the $L1$ distance loss function.

$$L_{cGAN}(G, D) = \mathbb{E}_{x \sim p_x(x)}[\log D(x, y)] + \mathbb{E}_{z \sim p_z(z)}[\log(1 - D(G(z, y)))], \tag{5}$$

$$L_d(G) = \mathbb{E}_{x,y,z}[\|y - G(x, z)\|_1]. \tag{6}$$

In PIX2PIX, the $L1$ distance loss function is used to measure the difference between the generated image and the real image, which can effectively preserve the high-frequency details of the image and make the generated image clearer. Additionally, a smaller distance loss, according to Equation (6), means that the generated result is closer to the true value, ensuring the consistency between the generated image and the real result and the unity of the style.

## 3. Proposed Method

### 3.1. Building Structure Inference Network

The overall process of the building structure inference network is shown in Figure 1. The method is divided into two parts: training and implying. During the training process, we extract features from UAV remote sensing images and corresponding CAD drawings to obtain a training set, including the network inputs and the corresponding inferred ground truth. We use PIX2PIX as the framework of the building structure inference network and introduce a specialized loss function for the additive attention gates. Through network training, we obtain a generator with inference capabilities. Its function is to infer the

internal layout of the building based on the exterior contour and the location of the entrance captured by the drone during the inference process, thus achieving the goal of this method.

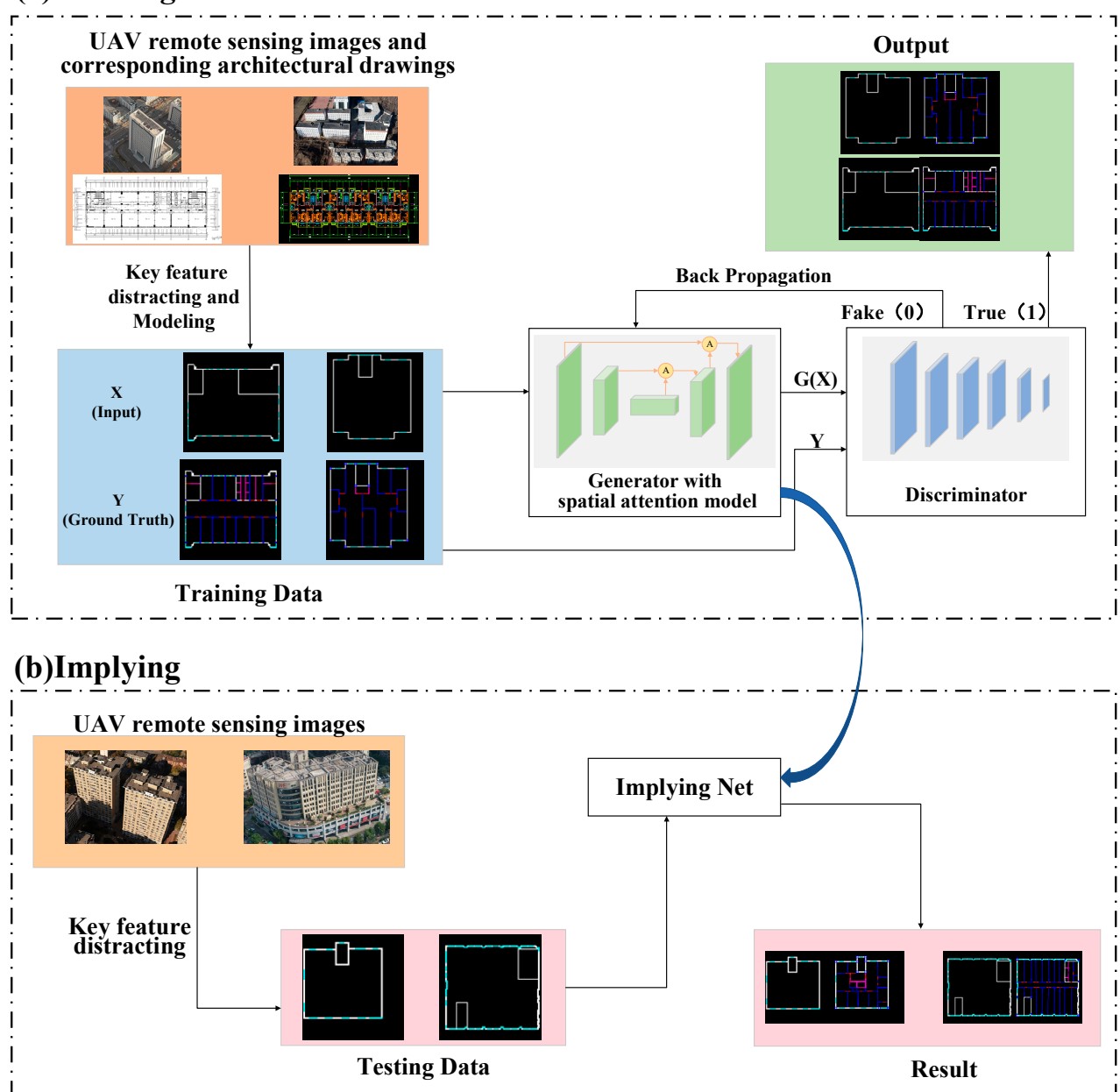

**Figure 1.** The overall process of the building structure inference network. (**a**) The process of training the building structure inference network. (**b**) The process of implying the building structure through UAV remote sensing images using the trained generator of the building structure inference network.

3.1.1. Generator

The generator of the building structure inference network uses U-net [32] as the basic framework. The U-net consists of symmetric downsampling and upsampling layers, and this symmetrical structure can be viewed as a learning/inference process, which is the key to the inference ability of our method, shown in Figure 2. During training, the network takes the exterior contour feature map of the building as the input. The encoder extracts its appearance features, and the decoder uses the extracted feature maps to perform inference and generate a building plan with the internal structure as the output of the

generator. Additionally, the U-net network adds skip connections between the encoding and decoding stages, connecting corresponding feature distributions. This allows the network to preserve detailed feature information at different scales, and the information can be directly transmitted through the skip connections, reducing the information loss during downsampling and ensuring the comprehensiveness of feature extraction.

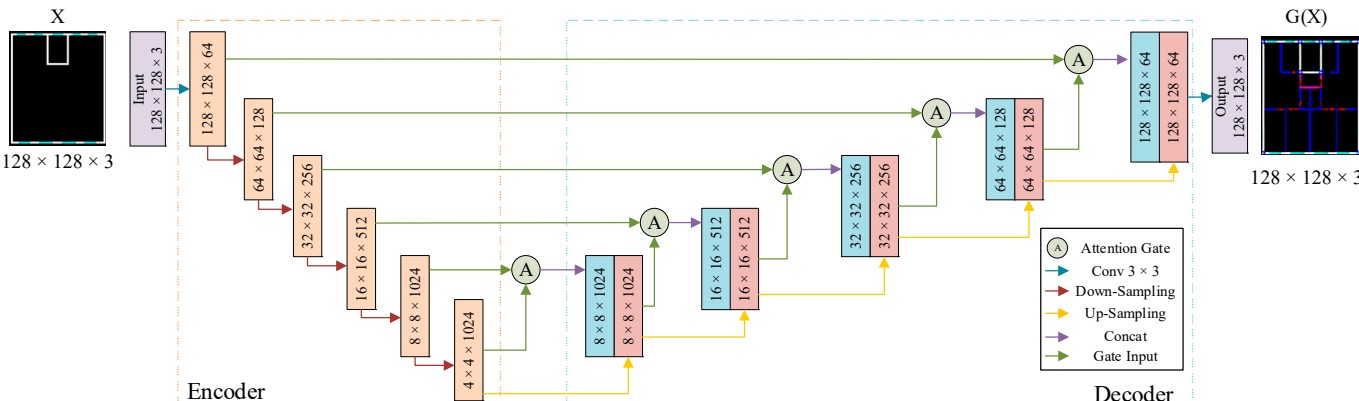

**Figure 2.** Composition of the generator of the building structure inference network.

In the inference of architectural structures, the internal structure of a building needs to be determined not only based on external features but also by considering the global relationship of the building, such as the layout of symmetric structures. However, in the downsampling process of the U-net network, the feature extraction process can lead to the partial loss of global information as the number of convolutional layers increases, which makes it difficult for the generator to reason about the global features of the building. In addition, according to the form of the dataset, the external contour features of the building and the layout of the floor plan are expressed in the form of lines, so the information extraction of the internal blank area is not the focus of the research. Therefore, the network needs to focus more on the relationships between lines when learning the rules to increase the attention on the regions of interest.

Considering the above two factors, we introduced an additive attention module that can enhance attention on regions of interest while maintaining the correlation of global features through the additive gate. This module improves the accuracy of the inference results while improving the calculation efficiency. The specific structure of the module is shown in the Figure 3.

In the original U-net architecture, the skip connections directly concatenate the feature maps of the encoding and decoding paths at the same spatial resolution. After introducing the attention module, as shown in Figure 3, the connection layer is modified to multiply the feature maps obtained through downsampling with the interest feature maps generated by the additive gating mechanism, which is represented by the attention map, as shown in Equations (7) and (8).

$$Q_a^l = C^T(\sigma_1(C^T x_i^l + C^T g_i + b_g)) + b_C, \tag{7}$$

$$\alpha_i^l = \sigma_2(Q_a^l(x_i^l, g_i; \theta_a)). \tag{8}$$

where $\sigma_1$ represents the ReLU activation function. $\sigma_2 = (x_{i,c}) = \frac{1}{1+\exp(-x_{i,c})}$ represents the Sigmoid activation function. $\theta_\alpha$ is a set of trainable parameters. $C$ is a $1 \times 1 \times 1$ convolutional layer that maps different feature maps into the same vector space. $b_g \in \mathbb{R}$ and $b_C \in \mathbb{R}^{F_{int}}$ represent different biases, which are set as the initial parameters of the attention module.

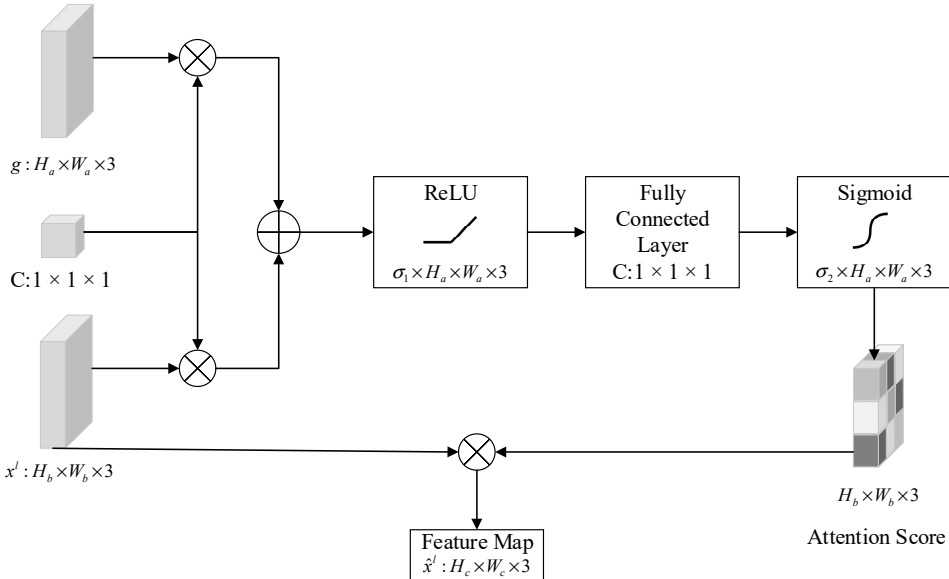

**Figure 3.** Structure of the attention gate.

During the upsampling process, the input of the attention module comes from different convolutional layers. For example, the input of the first attention module is the output feature maps of the last two layers of the encoder, while for the second attention module and beyond, the input is the output feature maps of the current decoder layer and the previous decoder layer. Therefore, this module links the features of adjacent layers together, as shown in Figure 3. Attention modules are incorporated into all five upsampling layers, which ensures the inter-correlation of features across the entire network, maintains the feature association from local to global, and reduces the loss of global features, thereby achieving cross-scale feature correlation.

Meanwhile, due to the fact that the RGB pixel values of the uninterested regions are [0,0,0], the attention scores of these regions are significantly reduced after a series of hidden layer computations in the additive gating mechanism. Therefore, after the multiplication of the feature maps, the feature values of these uninterested regions are small, which can help the network capture the features better.

Overall, the addition of the attention module helps the generator extract the regularity of building planar structures and reduces the impact of uninterested regions on feature extraction. At the same time, the attention module reduces the loss of global features during downsampling and strengthens global correlation, which has significant implications for building structure inference.

### 3.1.2. Discriminator

In the building structure inference network, the discriminator adopts a fully convolutional network and shares the same network skeleton as that used in PatchGAN [33]. The input to the discriminator is the generator's output and the ground truth image, and the features are extracted and compared by the fully convolutional network. In GANs, features are obtained by a multi-layer convolution, and the difference between the generated and ground truth images is compared to output a binary result of 0 or 1. A result of 1 denotes that the generated image conforms to the probability distribution of the ground truth, while 0 indicates that it does not meet the requirement. However, this binary classification method has limitations when dealing with complex generated results, such as those in this method for building structure inference. The convolution process can cause the disappearance of global features, resulting in a decrease in the discriminator's discriminative ability. When the generated results are not accurate enough, they are still classified as 1. Therefore, we need to adjust the structure of the discriminator to enhance its discriminative ability.

To address this issue, we chose PatchGAN as the discriminator skeleton. PatchGAN's output is not binary (0 or 1) but a matrix, with each element representing a part of the original image. This approach allows for a more detailed description of the generated results, thereby facilitating a better comparison between the generated and real images. The objective function is given as follows:

$$D^* = \text{argmax}\Big\{ E_{x,y \sim P_{data}(x,y)}[\log D(x,y)] + E_{x \sim P_{data}(x), z \sim P_z(z)}[\log(1 - D(x, G(x,z)))] \Big\}. \tag{9}$$

The network architecture of the discriminator is shown in Figure 4. After passing through four convolutional layers, a $30 \times 30$ feature matrix is obtained, with each element corresponding to a part of the original image, i.e., a patch. By comparing the two matrices, the discriminator can compare more features and thus has stronger discriminative ability.

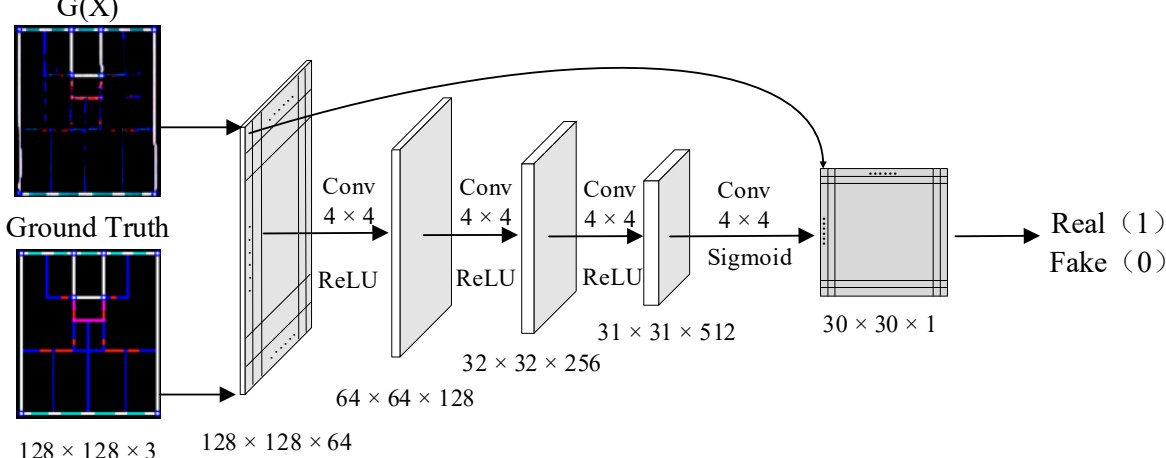

**Figure 4.** Composition of the discriminator of the building structure inference network.

### 3.2. Loss Function

Based on Equation (4), the loss function of the PIX2PIX network is mentioned above. However, in this task, using Equation (4) alone does not accurately represent the accuracy of the inference results. This is because the loss function of PIX2PIX places more emphasis on the overall image quality of the generated results. However, in this task, we need to better measure the accuracy of the inference results, including their accuracy and positional deviation. Therefore, we introduce Dice loss [34] into the loss function to quantify the degree of deviation between the inference results and the ground truth. Dice loss measures the overlap between the predicted and ground truth results and is particularly effective for pixel-level tasks. By combining Dice loss with the loss function of PIX2PIX, we obtain a more comprehensive and accurate loss function that can better measure the accuracy and positional deviation of the building structure inference and improve the quality of the inference results.

$$L_{Dice} = 1 - \frac{2|G(x) \cap y|}{|G(x)| + |y|}, \tag{10}$$

During the experiment, we found that the generated results often have incomplete straight elements, such as walls. To address this issue, we propose to introduce a penalty term to constrain the completeness of the walls. Considering that the Hough transform [35] is a commonly used method for detecting straight lines, we propose a penalty term based on a Hough transform to constrain the completeness of walls. The Hough transform detects straight lines by counting the number of intersection points between curves in the Hough space, which does not require the position of the straight lines in the image. Therefore, we can use the number of intersection points to constrain the completeness of the generated straight lines, which can effectively improve the quality of the generated results. This penalty term not only serves as a constraint but also helps to improve the expression of the loss function.

The derivation of the penalty term is shown below. First, the image is transformed into the Hough space, as shown in Equations (11) and (12).

$$\delta(I) = \{\forall(x_0, y_0) \in I | I(x_0, y_0) \to H(\rho, \theta)\}, \tag{11}$$

$$\rho = x_0 \cos\theta + y_0 \sin\theta, \tag{12}$$

Then, based on the values of $\rho$ and $\theta \in [0, 2\pi]$, we determine the curve corresponding to a certain point in the image space and search for the intersection points of the curve. The number of curves passing through the intersection point is denoted as $N(x_0, y_0)$. We then compare the value of the generated image, $N_{G(x)}(x_i, y_j)$, with that of the ground truth image, $N_y(x_i, y_j)$, for the same place. A smaller difference in the number of points indicates that the straight lines in the two images are closer together and more complete, as shown in Equation (13):

$$L_{Hough} = \frac{1}{M \times N} \times \sum_{\substack{i=1 \\ j=1}}^{M,N} \log\left| N_{G(x)}(x_i, y_j) - N_y(x_i, y_j) \right|, \tag{13}$$

According to the Hough transform, when the image is transformed into the Hough space, each point in the original image can be represented as a cluster of curves in the Hough space for different values of $\rho$ and $\theta$, and the intersection points of the curves represent the straight lines in the original image. Therefore, using the complete straight lines in the ground truth image as a reference, we compare the number of intersection points obtained after the Hough transform with that of the inferred results. If the numbers are close, it indicates that the inferred straight lines are relatively complete; if not, it indicates that the inferred results are not complete enough. To match the scale of the penalty term with the original loss function, we take the logarithm and calculate the mean of the penalty term, ensuring that all terms in the loss function are on the same scale. The process of the straight lines penalty term is shown as Algorithm 1.

---

**Algorithm 1.** Straight Lines Penalty Term

---

**Input**: Ground truth X and generator result Y
**Step 1.** Converting X and Y into binary images $X_b$ and $Y_b$.
**Step 2.** Transforming $X_b$ and $Y_b$ into the Hough domain.

- Select an arbitrary pixel point $p_i$ from $X_b$.
- Compute the calculation according to $r = x_i \cos\theta + y_i \sin\theta$, where the value of $\theta$ ranges from 0 to $2\pi$. $\theta \in [0, 2\pi]$
- Each pixel point corresponds to a curve r in the Hough domain. With m pixel points in $X_b$, there is a collection of curves $R_X$ including m curves.
  $R_X = \{r \in R : 0 \leq \theta \leq 2\pi, (x_i, y_i) \in X_b, r = x_i \cos\theta + y_i \sin\theta\}$
- Select an arbitrary pixel point $q_i$ from $Y_b$.
- Similarly, $R_Y$ can be computed.
  $R_Y = \{r \in R : 0 \leq \theta \leq 2\pi, (x_i, y_i) \in Y_b, r = x_i \cos\theta + y_i \sin\theta\}$

**Step 3.** Calculate the number of curves at the intersection point.

- Select an intersection point $g_i(r_i, \theta_i)$ from $R_X$. The number of the curves at $g_i$ is $N_X^i$.
- If $N_i \geq T$ (T means the threshold of the shortest line which can be detected), calculate the number of curves in $R_Y$ passing through point $g_i(r_i, \theta_i)$. The number of the curves at $g_i$ is $N_Y^i$.

**Step 4.** Calculate the straight lines penalty term

- Iterate through all the intersection points, then calculate the straight lines penalty term by
  $$L_{Hough} = \frac{1}{N} \sum_{i=1}^{N} \log\left| N_X^i - N_Y^i \right|$$

---

Therefore, the loss function, L, for the building structure inference network is:

$$L = \lambda_1 L_{PIX2PIX} + \lambda_2 L_{Dice} + \lambda_3 L_{Hough}, \tag{14}$$

where $L_{PIX2PIX}$ is defined as Equation (4), $L_{Dice}$ is defined as Equation (9), and $L_{Hough}$ is defined as Equation (12). $\lambda_1$, $\lambda_2$, and $\lambda_3$ are the weights assigned to each term to balance their contribution to the overall loss function.

By adjusting the weights, $\lambda_1$, $\lambda_2$, and $\lambda_3$, a more balanced result can be obtained, which ensures image clarity while also improving the accuracy and completeness of the inference, leading to an optimal result.

## 4. Experiments and Results

### 4.1. Dataset

Our method combines the exterior features of buildings, such as the layout positions of windows, balconies, entrances, and traffic cores, to infer their internal floor plan structures. However, existing datasets [36,37] suffer from the problem of unclear entrances and inadequate correspondence between exterior features and internal structure. To address the information requirements of building structure inference, we construct a dataset named RoomLayout based on CAD drawings of real buildings, including multi-story residential buildings, high-rise residential buildings, and multi-story office buildings. This dataset covers different building types with various functions and structural features, thereby improving the robustness of the network. The process of creating the training set is illustrated in Figure 5.

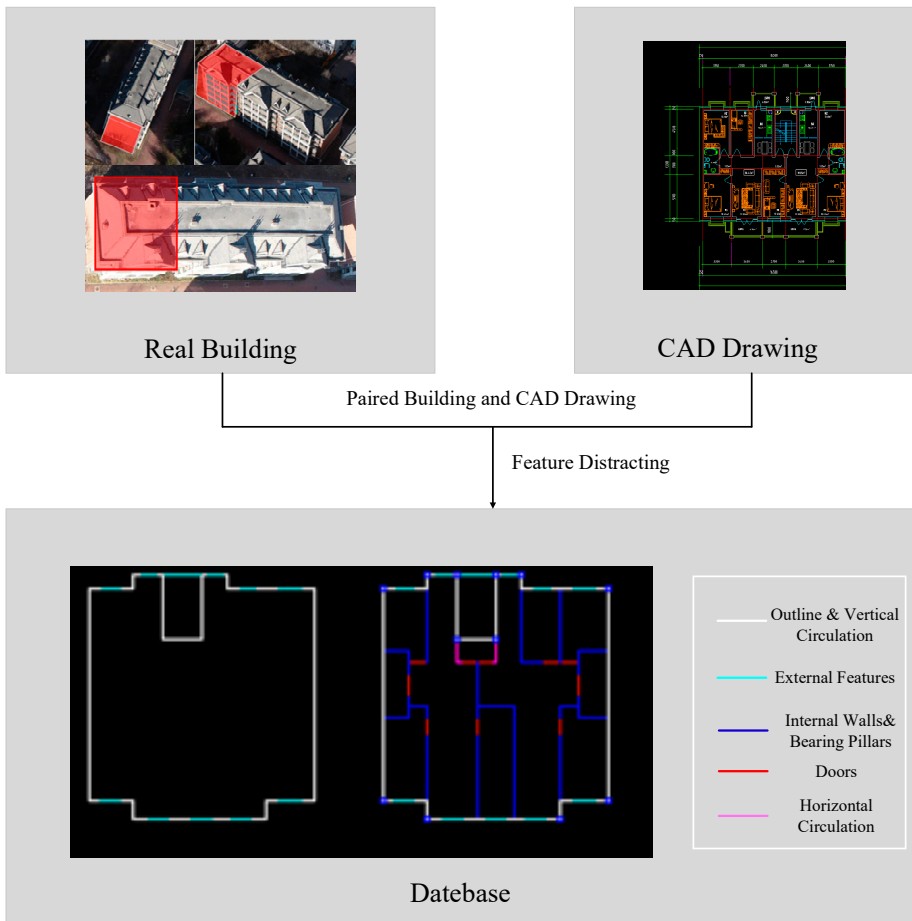

**Figure 5.** Process of making dataset.

Firstly, to produce the input for the generator in the network, the building's external outline and features, such as window and balcony positions and entrances, need to be extracted from the CAD floor plan. Based on expert knowledge of building structural design, it is known that the entrance of a building is connected to its internal vertical traffic core, which has a depth ranging from 1/2 to 1/3 of the building's overall depth. Therefore, the building's external outline and the position of its internal traffic core can be extracted as information for structural inference, which are used as inputs for the generator, as illustrated in the figure. During testing, the generator's outline is extracted from UAV remote sensing images, and the building's external features are further extracted accordingly. The input during testing is generated by following the drawing method used in the network's training dataset.

Secondly, the ground truth images are drawn. The training dataset for the building structure inference network is paired, so the internal floor layout of the building is drawn based on the CAD floor plan, and the load-bearing columns in the floor plan are specially marked. The purpose of marking the load-bearing columns during training is to discriminate them from other inference results in the final output, which helps to obtain a clearer result. As shown in the Figure 5, the white lines represent the external outline and the position of the vertical traffic core, the cyan color represents the external features, the blue color represents the position of the internal walls and load-bearing columns, the red color represents the doors, and the pink color represents the position of the horizontal traffic core.

The self-made RoomLayout dataset contains a total of 400 images, including 125 images of multi-story residential buildings, 134 images of high-rise residential buildings, and 116 images of office buildings. The data sources are real buildings and their corresponding CAD drawings. The size of the images is $256 \times 128$, with 320 images (4/5 of the total) used as the training set and 80 images (1/5 of the total) used as the test set. Some examples from the dataset are shown in the Figure 6.

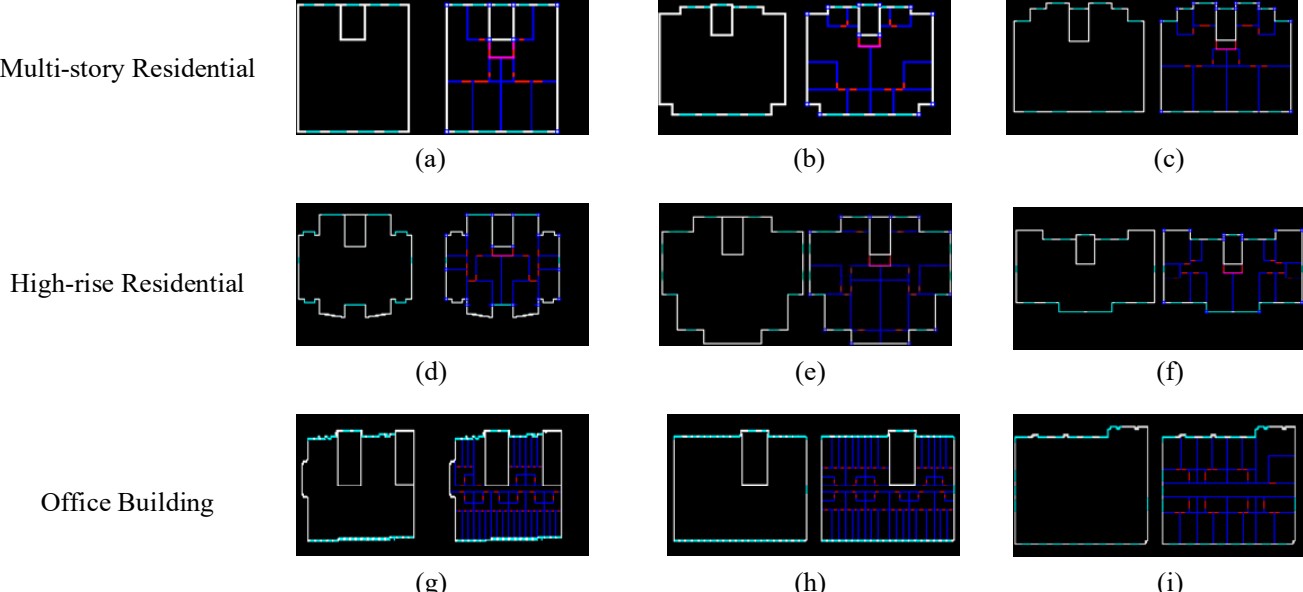

**Figure 6.** Examples of three kinds of building in the dataset. (**a**–**c**) are multi-story residential buildings. (**d**–**f**) are high-rise residential buildings. (**g**–**i**) are office buildings.

*4.2. Evaluation Metrics*

In this experiment, two metrics were used to evaluate the inference results, namely Root Mean Square Error (RMSE) and Intersection over Union (IoU) [23]. The calculation formulas for these two metrics are shown below:

$$d_{\text{RMSE}} = \sqrt{\sum_{i=1}^{N} (\Delta X_i{}^2 + \Delta Y_i{}^2)/N}, \tag{15}$$

$$d_{\text{IoU}}(X, Y) = \sum_{i=1}^{n} \left(\frac{x_i \cap y_i}{x_i \cup y_i}\right) \times \frac{1}{n}, \tag{16}$$

where RMSE measures the degree of layout displacement in the inference results by calculating the deviation distance between each pixel in the inference results and its corresponding pixel in the ground truth. $N$ is the total number of pixels in the inference results. IoU evaluates the accuracy of the inference results by calculating the overlapping degree between the walls in the inference results and those in the ground truth. $n$ is the total number of walls.

*4.3. Experiment Settings*

The experiment was conducted on the Python 3.7 platform with PyTorch 1.13.1. The hardware used for the experiment was an NVIDIA RTX3060 GPU and a 12th Gen Intel(R) Core(TM) i7-12700 CPU. According to Equation (13), the weights of the loss function were set to $\lambda_1 = 0.3$, $\lambda_2 = 0.2$, and $\lambda_3 = 0.5$. In the training parameter settings, we set the epoch to 2000 and the batch size to 4. The learning rates for the generator and discriminator were set to 0.0002 and 0.0005, respectively. This learning rate setting was designed to better train the generator and prevent the discriminator from becoming too strong due to differences in the complexity of the generator and discriminator networks. The weight of the penalty term was maximized as much as possible while ensuring that the generator and discriminator converged easily in order to ensure the integrity of the walls in the inference results.

*4.4. Results and Analysis*

From Figure 7, we can see that the cGAN network, which uses a CNN-based discriminator, has lower image generation quality and accuracy compared to the PIX2PIX network with a PatchGAN discriminator and *L*1 loss function. Furthermore, our proposed method, which incorporates attention modules to enhance the correlation of global features and designs a specialized loss function to constrain the integrity of walls, achieves more accurate inference results than PIX2PIX method. Our method performs well on both residential buildings with relatively symmetrical room layouts and office buildings with more complex interiors.

Comparing the experimental results, we can see that our method achieves higher accuracy in inferring symmetrical building structures. This is because in the process of extracting regularities, symmetry information relies more on the extraction of global features, which has a better extraction effect after the introduction of attention modules. Therefore, compared with the PIX2PIX method, our method achieves higher accuracy in inferring symmetrical structural layouts, which confirms the feasibility of attention modules.

Based on Table 1, it is evident that our method has a better performance than the compared method in all three categories of multi-story residential buildings, high-rise residential buildings, and office buildings. RMSE measures the deviation between the inferred results and the ground truth in pixels, and the spatial resolution of the image is determined by the exterior contour scale of the building. By converting the building size, we can estimate that the deviation for residential buildings is approximately between 350 and 550 mm, and for office buildings, it is approximately between 500 and 600 mm. Compared to PIX2PIX, our method achieves a 12.3%, 13.6%, and 7.6% improvement in multi-story

residential buildings, high-rise residential buildings, and office buildings, respectively. Moreover, the IoU results demonstrate that our method outperforms PIX2PIX by 21.9%, 20.2%, and 21.2% in multi-story residential buildings, high-rise residential buildings, and office buildings, respectively. This is because most residential buildings have symmetrical structures and fewer internal inference elements compared to office buildings, resulting in higher IoU for both types of residential buildings as compared to office buildings.

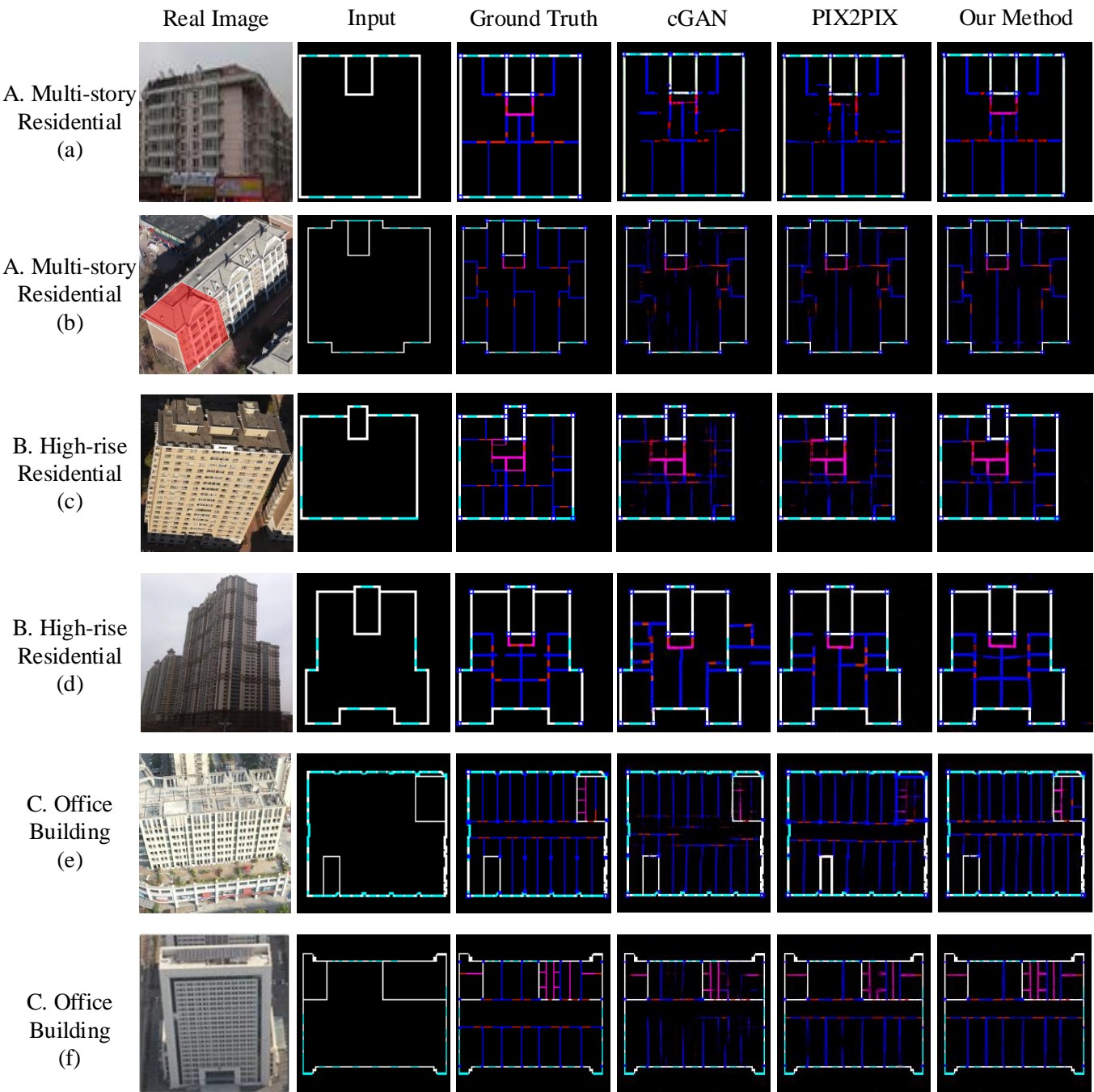

**Figure 7.** Results of building structure inference network.

**Table 1.** Evaluation indicators for different types of buildings.

| | Multi-Story Residential | | High-Rise Residential | | Office Building | |
|---|---|---|---|---|---|---|
| | **RMSE (Pixel)** | **IoU** | **RMSE (Pixel)** | **IoU** | **RMSE (Pixel)** | **IoU** |
| cGAN | $2.093 \pm 0.214$ | $0.525 \pm 0.066$ | $2.017 \pm 0.196$ | $0.549 \pm 0.062$ | $1.708 \pm 0.202$ | $0.408 \pm 0.059$ |
| PIX2PIX | $1.461 \pm 0.158$ | $0.682 \pm 0.054$ | $1.433 \pm 0.147$ | $0.677 \pm 0.058$ | $1.115 \pm 0.136$ | $0.495 \pm 0.050$ |
| Our method | $\mathbf{1.281 \pm 0.135}$ | $\mathbf{0.832 \pm 0.051}$ | $\mathbf{1.238 \pm 0.129}$ | $\mathbf{0.814 \pm 0.049}$ | $\mathbf{1.030 \pm 0.095}$ | $\mathbf{0.682 \pm 0.051}$ |

By calculating the trainable parameters of the network, we can estimate the complexity of the method. According to Table 2, our method has more trainable parameters compared to PIX2PIX and cGAN. When running for 2000 epochs with the same batch size and learning rate, our method takes a longer time. However, our method achieves higher accuracy in the task of building structure inference. From the results of the inference, the investment in model complexity and training time is justified.

**Table 2.** Network parameters (params) and the training time of different methods.

| Method | Params | Training Time (min) |
|---|---|---|
| cGAN | 25.7 M | 19.3 |
| PIX2PIX | 44.6 M | 44.1 |
| Our method | 101.1 M | 129.6 |

From Table 3 and Figure 8, it can be observed that after introducing the Dice loss function, the improvement in IoU for the three types of buildings were 5.9%, 9.1%, and 24.9%, while the improvement in RMSE were 1.6%, 1.6%, and 10.0%, respectively. The improvement in IoU is more significant than that in RMSE because the Dice loss function also measures the overlap between the inferred and ground truth results. The reason for the more significant improvement in IoU for office buildings compared to residential buildings is that the internal layout of office buildings is more uniform, allowing for better inference of repeated elements after introducing the Dice loss function. After introducing the Hough domain-based penalty term, the improvement in RMSE for residential buildings was more pronounced, with an increase of 8.7% and 12.5%. This is because the penalty term emphasizes the integrity of the walls, which leads to an increase in the number of pixels in the walls and a decrease in the RMSE, resulting in better results. From the results in Figure 8, it is evident that after introducing the penalty term, the completeness of the walls has improved compared to the PIX2PIX method, indicating that the loss function we designed has the ability to constrain the completeness of the walls.

**Table 3.** Evaluation indicators for different types of loss.

| | Multi-Story Residential | | High-Rise Residential | | Office Building | |
|---|---|---|---|---|---|---|
| | **RMSE (Pixel)** | **IoU** | **RMSE (Pixel)** | **IoU** | **RMSE (Pixel)** | **IoU** |
| $L_{PIX2PIX}$ | $1.426 \pm 0.145$ | $0.707 \pm 0.069$ | $1.415 \pm 0.139$ | $0.692 \pm 0.065$ | $1.096 \pm 0.098$ | $0.513 \pm 0.050$ |
| $L_{PIX2PIX} + L_{Dice}$ | $1.403 \pm 0.143$ | $0.749 \pm 0.072$ | $1.392 \pm 0.136$ | $0.755 \pm 0.068$ | $\mathbf{0.986 \pm 0.090}$ | $0.641 \pm 0.061$ |
| $L_{PIX2PIX} + L_{Dice} + L_{Hough}$ | $\mathbf{1.281 \pm 0.135}$ | $\mathbf{0.832 \pm 0.051}$ | $\mathbf{1.218 \pm 0.129}$ | $\mathbf{0.814 \pm 0.049}$ | $1.030 \pm 0.095$ | $\mathbf{0.682 \pm 0.051}$ |

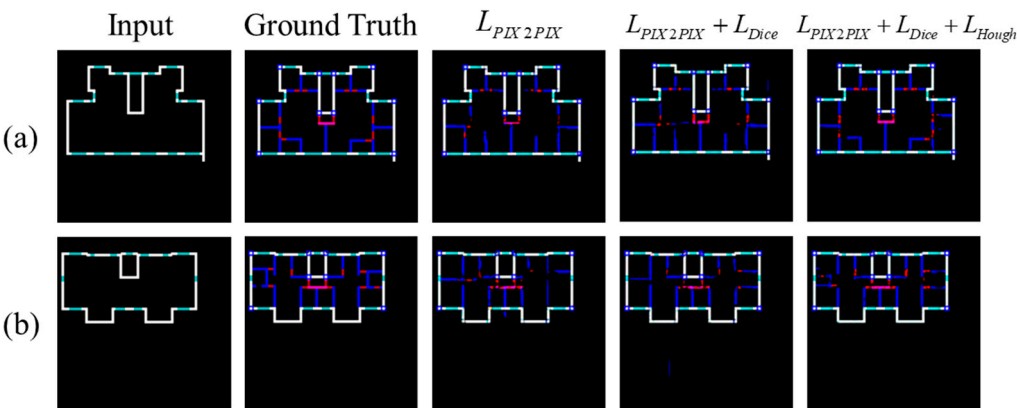

**Figure 8.** Results of building structure inference network with different loss functions. (**a**,**b**) are two examples of the building inference network with different loss functions.

## 5. Conclusions and Discussion

In this paper we propose a deep learning method for building structure inference based on the external key information of buildings, aiming to address the lack of deep learning methods for building structure inference. Our method extracts the building outline and external features from remote sensing images obtained by drones and uses a building structure inference network based on the PIX2PIX network as the backbone. An additive attention module is introduced in the generator, which not only improves computational efficiency but also effectively extracts global features and enhances the inference capability of the network. In addition, a new loss function is proposed, which introduces the Dice loss and a penalty term based on the Hough domain combined with the PIX2PIX loss function. The Dice loss is used to optimize the accuracy of inference, while the penalty term emphasizes the integrity of straight-line elements in the inference results. The experimental results show that the proposed method has high accuracy in inferring the structures of three types of buildings: multi-story residential, high-rise residential, and office buildings. The deviation of the inferred results is within 600 mm after conversion, and the highest IoU of the images can reach 0.832, demonstrating the potential of the proposed method as an automated solution for building structure inference.

Although the building structure inference network can accurately infer the building structure based on UAV images, there are still some limitations to this method. Firstly, it does not perform well in inferring complex architectural structures. Since the dataset is manually made, the data volume is not large enough to cover all architectural design patterns comprehensively. Additionally, the majority of samples in the dataset are residential and office buildings, requiring a greater diversity of complex architectural samples in the dataset. Secondly, the computational speed is limited. While our method provides more accurate results compared to existing methods such as CGAN and PIX2PIX, the training time of the model remains a significant challenge. Therefore, further research is needed to simplify the model and reduce its computational complexity while maintaining its accuracy.

**Author Contributions:** Conceptualization, H.C.; Methodology, Z.G.; Software, Z.G.; Formal analysis, Z.G.; Investigation, F.H.; Writing—original draft, Z.G.; Writing—review & editing, Z.G. and F.H.; Visualization, F.H.; Project administration, H.C.; Funding acquisition, H.C. and X.M. All authors have read and agreed to the published version of the manuscript.

**Funding:** This research received no external funding.

**Data Availability Statement:** Not applicable.

**Conflicts of Interest:** The authors declare no conflict of interest.

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
