# Peer review of "A Generative Adversarial Network with Spatial Attention Mechanism for Building Structure Inference Based on Unmanned Aerial Vehicle Remote Sensing Images"

_remotesensing, doi:10.3390/rs15184390_

Round 1

Reviewer 1 Report

The authors of this paper presented a generative adversarial network with a spatial attention mechanism based on UAV remote sensing images to infer buildings’ internal structures. A dataset of the appearance feature obtained from UAV remote sensing images and the internal floor plan structure is first made. To establish the mapping relationship from the dataset through the external appearance with the internal structure of the buildings, the PIX2PIX network is used as the basic framework of our method. An additive attention module is added to the generator, which uses multi-scale feature fusion to combine features from different spatial resolutions of the feature map, improving the model's ability to focus on global relationships in the mapping. To ensure the completeness of line elements in the generator’s output, we design a novel loss function based on the Hough transform. A line penalty term is introduced, which transforms the output of the generator and ground truth to the Hough domain due to the original loss function's inability to effectively constrain the completeness of straight-line elements in the generated results in the spatial domain, then minimize the difference between the intersection points to measure the completeness of the lines. The paper is well-presented, and the topic is good. It can be accepted for publication with a major revision as follows:

- Improve the presentation of the abstract. Focus on your main contribution.

- The quality of some figures must be improved.

- More details about the proposed method are needed, So pseudocodes or real codes can be added. You can add the pseudocodes to the main text of this paper. And you can share your source codes on a public platform.

- Recheck the caption of Figure 8;

- Complexity and time computation can be discussed.

- Limitations and challenges must be highlighted.

- English Proofreading is needed.

- Related work must include recent studies, such as: UAV Aerial Image Generation of Crucial Components of High-Voltage Transmission Lines Based on Multi-Level Generative Adversarial NetworkI; DA: Improving distribution analysis for reducing data complexity and dimensionality in hyperspectral images; 

Moderate editing is needed 

Author Response

We feel great thanks for your professional review work on our manuscript entitled “A GAN with Spatial Attention Mechanism for Building Structure Inference Based on UAV Remote Sensing Images” (Manuscript ID remotesensing-2508787). As you are concerned, there are several problems that need to be addressed. According to your nice suggestions, we have made extensive corrections to our previous draft in the attachment PDF.

Reviewer 2 Report

With the process of urbanization, some older buildings need to be demolished. This paper puts forward a method to solve this problem by using UAV, which has potential significance and value. However, in the actual demolition, there are still many problems that need to be solved, such as the distinction of load-bearing walls and so on. These more professional problems are practical applications will face the problem. However, this does not affect the integrity of this article. The structure of this paper is complete and recommended.

Author Response

Thank you very much for taking valuable time to review our manuscript and for your valuable feedback. We sincerely appreciate your recognition of our work.

Reviewer 3 Report

Review Summary:
Title: A GAN with Spatial Attention Mechanism for Building Structure Inference Based on UAV Remote Sensing Images Author(s): Hao Chen, Zhixiang Guo, Xing Meng, and Fachuan He
Decision: Not Accepted in its Present Form
Main Reasons for Decision:
1. Lack of Significant Advancement: Upon thorough evaluation of the content presented in this paper, it becomes evident that the net new advancement brought forth is not significant. For a journal of the standing of "Remote Sensing," it is imperative that submitted works provide notable contributions to the existing body of knowledge. In its current form, this paper does not seem to meet this criterion.
2. Over-reliance on Established Models: Most of the paper describes the PIX2PIX and PatchGAN models. While these are undoubtedly well-known and highly cited models in GAN, merely describing them without offering substantial novelty or a new perspective does not make a compelling case for publication.
3. Insufficient Scientific Rigor: A fundamental expectation for any submission to "Remote Sensing" is that the paper maintains a high degree of scientific rigor. This entails not just sound methodology but also a comprehensive discussion, critical analysis, and a systematic approach to the topic at hand. Unfortunately, this paper falls short of sounding sufficiently scientific. The arguments are not cogently structured, and the narrative lacks the depth expected from a publication in this domain.
I have some broad remarks about the present draft:
1. Abstract: An abstract should succinctly highlight the paper's key innovations, distinguishing it from prior research. By emphasizing these novelties, the abstract showcases how the study advances scientific knowledge, capturing the interest of readers seeking the latest breakthroughs in the field. I would suggest authors rewrite this section by highlighting key advancements decribed in the article. 2. L112: Adding explanation on why existing LOSS function cannot optimize the model would help readers understand the motivation behind this research. 3. L253: It is unclear to me why adding attention layers without modifying rest of the NN architecture would increase computation efficiency 4. L324: I would advice authors to discuss some limitations of the new LOSS function as it increases complexity to the model training 5. L366: I could be wrong, but only 400 images might be too small of a sample size to train a large NN model without the risk of overfitting. I would like to see authors consider the risk of such a low sample volume.' 6. All example plots of the dataset only captures rectangular structures. I wonder if authors thought about limitation of current model when coming across more complex building architechtures 7. Table 1, Table 2: Adding confidence interval would help readers get a sense of improvement through new model authors presented
While the current form of the paper is not suitable for publication in "Remote Sensing", the effort put into the work is acknowledged. It is recommended that the authors consider refining their approach, emphasizing unique contributions, and bolstering the scientific narrative.

NA

Author Response

We feel great thanks for your professional review work on our manuscript entitled “A GAN with Spatial Attention Mechanism for Building Structure Inference Based on UAV Remote Sensing Images” (Manuscript ID remotesensing-2508787). After carefully reviewing your main reasons for the decision, we believe that the innovation of our work can be primarily attributed to two aspects. Firstly, through our investigation, we observed a scarcity of research focusing on inferring internal structures of buildings using aerial imagery captured by UAVs. Consequently, we conducted research on intelligent inference of building structures based on deep learning networks, which represents a novel contribution. Secondly, while leveraging PIX2PIX as the foundational framework for our network, we incorporated an attention mechanism and introduced new loss functions to tailor the approach specifically for the task at hand. Particularly noteworthy is the inclusion of a line penalty term based on the Hough transform, which enhances inference accuracy. Additionally, we employed UAVs to capture images of buildings in Changsha, Hunan Province, and Harbin, Heilongjiang Province, China, and inputted them into the network for inference, yielding results of practical significance. As you concerned, there are several problems that need to be addressed. According to your nice suggestions, we have made extensive corrections to our previous draft, the detailed corrections are in the attachment PDF.

Round 2

Reviewer 1 Report

The authors have addressed all comments, and this version can be considered for publication.

Reviewer 3 Report

I think the authors have answered most of my questions and updated the draft sufficiently to approve for acceptance.